# *Sphingomonas* and *Phenylobacterium* as Major Microbiota in Thymic Epithelial Tumors

**DOI:** 10.3390/jpm11111092

**Published:** 2021-10-26

**Authors:** Rumi Higuchi, Taichiro Goto, Yosuke Hirotsu, Sotaro Otake, Toshio Oyama, Kenji Amemiya, Hiroshi Ohyama, Hitoshi Mochizuki, Masao Omata

**Affiliations:** 1Lung Cancer and Respiratory Disease Center, Yamanashi Central Hospital, Yamanashi 400-8506, Japan; r-higuchi1504@ych.pref.yamanashi.jp (R.H.); ootake.sotaro.gx@mail.hosp.go.jp (S.O.); 2Genome Analysis Center, Yamanashi Central Hospital, Yamanashi 400-8506, Japan; hirotsu-bdyu@ych.pref.yamanashi.jp (Y.H.); amemiya-bdcd@ych.pref.yamanashi.jp (K.A.); ooyama-bdcx@ych.pref.yamanashi.jp (H.O.); h-mochiduki2a@ych.pref.yamanashi.jp (H.M.); m-omata0901@ych.pref.yamanashi.jp (M.O.); 3Department of Pathology, Yamanashi Central Hospital, Yamanashi 400-8506, Japan; t-oyama@ych.pref.yamanashi.jp; 4Department of Gastroenterology, The University of Tokyo Hospital, Tokyo 113-8655, Japan

**Keywords:** thymoma, microbiome, 16S RNA sequencing, genera, driver mutation

## Abstract

**Simple Summary:**

In this study, we evaluated the microbiota in resected thymoma samples and identified *Sphingomonas* and *Phenylobacterium* as the dominant genera in thymomas. This is the first study that evaluated the microbiota in thymoma and that identified bacterial genera specific to thymoma. Furthermore, our study indicates a potential approach for preventing the development of thymoma as a new “precision medicine”.

**Abstract:**

The microbiota has been reported to be closely associated with carcinogenesis and cancer progression. However, its involvement in the pathology of thymoma remains unknown. In this study, we aimed to identify thymoma-specific microbiota using resected thymoma samples. Nineteen thymoma tissue samples were analyzed through polymerase chain reaction amplification and 16S rRNA gene sequencing. The subjects were grouped according to histology, driver mutation status in the *GTF2I* gene, PD-L1 status, and smoking habits. To identify the taxa composition of each sample, the operational taxonomic units (OTUs) were classified on the effective tags with 97% identity. The Shannon Index of the 97% identity OTUs was calculated to evaluate the alpha diversity. The linear discriminant analysis effect size (LEfSe) method was used to compare the relative abundances of all the bacterial taxa. We identified 107 OTUs in the tumor tissues, which were classified into 26 genera. *Sphingomonas* and *Phenylobacterium* were identified as abundant genera in almost all the samples. No significant difference was determined in the alpha diversity within these groups; however, type A thymoma tended to exhibit a higher bacterial diversity than type B thymoma. Through the LEfSe analysis, we identified the following differentially abundant taxa: Bacilli, Firmicutes, and Lactobacillales in type A thymoma; Proteobacteria in type B thymoma; Gammaproteobacteria in tumors harboring the *GTF2I* mutation; and Alphaproteobacteria in tumors without the *GTF2I* mutation. In conclusion, *Sphingomonas* and *Phenylobacterium* were identified as dominant genera in thymic epithelial tumors. These genera appear to comprise the thymoma-specific microbiota.

## 1. Introduction

Early microbiome research focused primarily on gastrointestinal diseases, such as pseudomembranous enterocolitis, inflammatory bowel disease, and irritable colitis [1]. Recently, the human intestinal microbiota has been reported to be involved in carcinogenesis and cancer progression, and this phenomenon has been attracting attention [2,3]. In addition, the microbiota has been identified in tissues of the pancreatic, lung, and breast cancers through advanced sequencing technology [4,5,6,7,8].

Thymoma is a relatively rare mediastinal tumor with malignant potential that is difficult to treat [9,10]. According to the histological classification by the World Health Organization, thymomas can be categorized into types A, AB, B1, B2, and B3, depending on the tumor cell morphology and proportion of coexisting lymphocytes [11]. Type A thymomas are the least aggressive with the best prognosis; the extent of the aggressiveness increases and the prognosis worsens according to the following order: type A, AB, B1, B2, and B3 [12,13]. Thymoma has been reported to commonly occur in people aged 40–60 years [14]. The development of thymoma is not associated with smoking habits or sex; its causes are unknown [15]. However, thymoma coexists in approximately 20% of patients with myasthenia gravis [16,17]. Owing to the absence of an effective treatment other than surgical resection, there is an urgent need to elucidate the pathology and establish preventive measures and new treatment strategies for thymoma [18,19,20,21].

Although recent reports have indicated an association between the microbiome and colorectal, oral, pancreatic, lung, and other cancers [22,23,24,25,26,27], there is no report on the microbiome in thymoma. Unlike oral, gastrointestinal, and respiratory cancers, which have been previously reported, thymoma is anatomically located in the anterior mediastinum, and it does not communicate with the outer environment. Since the tumor environment of thymoma has been theoretically assumed to be sterile, microbiome research has not been conducted in the past. Consequently, little progress has been made in research on the involvement of the microbiota in the pathology of thymoma.

In this study, we performed a polymerase chain reaction (PCR) to amplify the 16S ribosomal RNA (rRNA) region in the bacterial genome in resected thymoma samples. Subsequently, we performed 16S rRNA sequencing and metagenomic analyses using next-generation sequencing to investigate the composition and diversity of the microbiota and to identify thymoma-specific microbiota. On the basis of the results from these analyses, we presented a predictive model of pathogenesis and evaluated its potential for the prevention and control of the development of thymoma.

## 2. Results

### 2.1. Patient Characteristics

Nineteen consecutive patients with thymomas who underwent surgery at our hospital between January 2014 and August 2020 were enrolled without bias. In three patients with type AB thymomas, the type A and B portions were microdissected and examined separately. Thus, in total, 22 tissue samples were analyzed for microbiota. The clinicopathologic characteristics of the patients are summarized in Table 1. The 19 patients were divided into groups by the following characteristics: 11 males, 8 females; 12 smokers, 7 nonsmokers; histological type A (five), AB (three), B1 (five), B2 (four), or B3 (two); and Masaoka stage I (seven), II (nine), III (two), or IV (one). The diameter of the tumor was between 20 and 95 mm, with a mean tumor diameter of 43.6 ± 22.8 mm. Patients’ ages at the time of surgery were between 42 and 81 years (68.2 ± 12.9 years). One patient with type B2 thymoma (Case 20; Figure 1) had a comorbidity of myasthenia gravis. 

### 2.2. OTU Analyses

A total of 136 OTUs were identified in the 22 samples, while no OTU was found in the negative control samples (without any tissue). The dominant (>1% average relative abundance) classifiable OTUs belonged to four families: namely, Sphingomonadaceae (abundance: 62.0 ± 12.2%), Caulobacteraceae (abundance: 23.9 ± 7.6%), Bradyrhizobiaceae (abundance: 6.7% ± 5.3%), and Phyllobacteriaceae (abundance: 2.0% ± 1.4%) (Appendix A). We identified 107 genera (>1% average relative abundance); the predominant genera are presented in Figure 1. The top two genera with a high abundance and composition were *Sphingomonas* (abundance: 66.9 ± 10.8%) and *Phenylobacterium* (abundance: 26.0 ± 8.7%). *Sphingomonas* was detected in all the samples, and *Phenylobacterium* was detected in all the samples except in case 18. Both bacterial genera were significantly more abundant than the others (Appendix A).

### 2.3. Differences in Microbiota between Thymomas and Pancreatic Cancers

To identify the thymoma-specific microbiota, we compared the microbiota between thymoma and pancreatic cancer (Appendix A). In the thymoma samples, compared with the pancreatic cancer samples, *Phenylobacterium, Phyllobacterium,* and *Sphingomonas* were significantly more abundant (Figure 2). Since *Phenylobacterium*, *Phyllobacterium*, and *Sphingomonas* were detected only in four, three, and eight of the 30 pancreatic cancer samples, respectively, the composition of these genera in the 22 thymoma samples (detected in 21, 18, and 22 samples, respectively) was significantly higher.

### 2.4. Analysis of Microbial Diversity within Groups

The Shannon Index was calculated to evaluate the bacterial diversity within the different groups. No significant differences were observed in terms of the histology, presence or absence of the *GTF2I* mutation, PD-L1 expression, and smoking habits (Figure 3). However, the type A samples exhibited a tendency toward increased microbiome diversity compared with the type B samples (*p* = 0.059, Figure 3A).

### 2.5. Analysis of Differentially Abundant Taxa

To further identify the specific species in every group, we used the LEfSe method to identify the differentially abundant taxa at each level. First, in the type A and B histological groups, we identified four differential bacterial taxa, including two phyla, Firmicutes and Proteobacteria; one class, Bacilli; and one order, Lactobacillales (Figure 4A). The differential features were Firmicutes, Bacilli, and Lactobacillales in type A thymomas and Proteobacteria in type B thymomas (Figure 4B). Alphaproteobacteria was dominant in thymomas without the *GTF2I* mutation, while Gammaproteobacteria was dominant in thymomas harboring the *GTF2I* mutation (Figure 4C,D). No differential bacterial composition and abundance were observed in association with the stage, PD-L1 expression, or smoking habits.

## 3. Discussion

In this study, the sequencing of microbiota in resected thymoma samples identified two genera, *Sphingomonas* and *Phenylobacterium*, in almost all the thymoma samples; the bacterial composition and abundance of these genera were markedly high. We separately analyzed type AB thymoma for type A and type B components and detected *Sphingomonas* and *Phenylobacterium* in both components. Although the oral microbiome is likely to affect and contaminate the lung microbiome, thymoma is anatomically unlikely to be affected by the oral microbiome [28,29]. The composition and abundance of these two genera were significantly higher in the microbiota of thymoma tissues than in the microbiota of pancreatic cancer tissues. Our results suggest that these two genera are thymoma-specific microbiota. In addition, we chose pancreatic cancer as control because the pancreatic cancer and thymoma tissue samples were analyzed in the same process at the genome analysis center of our hospital during the same period. There is also a factor common to both the pancreatic cancer and thymoma: they do not communicate directly with outer environment. This analysis suggested that the presence of the two genera was not a result of contamination during the analysis process. In contrast, *Sphingomonas* and *Phenylobacterium* have not been detected in lung cancer tissues according to recent reviews on the microbiota in patients with lung cancer [4,30,31,32]. In this study, because these two genera were detected in almost all the thymoma samples, it was suggested that *Sphingomonas* and *Phenylobacterium* may represent differential microbiome functions in thymoma development.

*Sphingomonas* is a bacterial genus that was subclassified from *Pseudomonas* approximately 30 years ago. Members of the former are Gram-negative bacteria; however, they do not contain lipopolysaccharides specific to Gram-negative bacteria [33]. Instead, these bacteria contain glycosphingolipids, which are found in eukaryotic cells [33]. They are common microorganisms inhabiting various environments, such as water environments (e.g., freshwater and seawater), soil, and plant root systems. The wide ecological distribution of these bacteria is attributed to their ability to use diverse organic compounds and their strong vitality, allowing them to survive in nutrient-poor environments [34]. Although several bacteria in the genus *Sphingomonas* were isolated in relatively clean environments, certain bacterial species were isolated in contaminated environments containing toxic organic compounds, such as polychlorinated biphenyl, creosote, and pentachlorophenol [35]. Subsequent studies revealed that these bacteria take up certain organic contaminants and use them as energy sources [36]. On the basis of these findings, progress has been made in elucidating the mechanism through which *Sphingomonas* metabolizes organic contaminants. Furthermore, several attempts have been made worldwide for applying this mechanism in environmental cleanup (bioremediation). Meanwhile, with respect to the microbiome, *Sphingomonas* has been reported to be enriched as blood microbiota in the serum of healthy patients and patients with breast cancer who exhibit a favorable prognosis [37,38].

Species within the genus *Phenylobacterium* are capable of degrading xenobiotic compounds with a phenyl moiety such as chloridazon, antipyrine, pyramidon, or their analogs [39]. Additionally, these bacteria can degrade polycyclic aromatic hydrocarbons [40]. *Phenylobacterium* has now been used in the bioremediation of a petroleum-contaminated soil to degrade polycyclic aromatic hydrocarbons and their analogs [41]. Unlike *Sphingomonas*, there has been no report of the detection of *Phenylobacterium* as blood microbiota. Future studies are expected to elucidate how *Sphingomonas* and *Phenylobacterium*, which are two genera of environmentally indigenous bacteria used for bioremediation, coexist in thymoma and how they are involved in the carcinogenic mechanism of thymoma.

Several indigenous microorganisms exist in the epithelium of the whole human body (e.g., the mouth, ear, nasal cavity, respiratory organs, digestive tract, skin, and reproductive organs); form microbiota; play various roles in the body; and form a symbiotic relationship with humans [1,2]. In recent years, it has been considered that disturbance in the microbiota composition (dysbiosis) may alter the risk of disease development, and there is a growing number of reports on the association between intestinal microbiota and several diseases, such as allergy, cancer, multiple sclerosis, Parkinson’s disease, depression, inflammatory bowel disease, and rheumatism [30]. Furthermore, sterilization and specific pathogen-free breeding have been reported to alleviate or cure these diseases in pathological mouse models [42]. Improvement of the microbiota may additionally prevent the development of diseases in humans [43]. If one or several species of bacteria cause a disease, they can be potential therapeutic targets. For example, the eradication of *Helicobacter pylori* is the standard of care for the prevention of gastric cancer in infected patients at present [44]. This study indicates that the microbiota may be associated with thymoma. The clinical application of this finding may pave the way for the prevention of thymoma through controlling the growth of the bacterial genera *Sphingomonas* and *Phenylobacterium*. Patients with myasthenia gravis are at a high risk of developing thymoma [16,17], and the prevention of thymoma is important for their long-term survival. In this study, case 20 involved a patient with thymoma complicated by myasthenia gravis (Figure 1); this patient was positive for *Sphingomonas* and *Phenylobacterium*, which were abundant. The development of probiotic models for antibiotics, vaccines, and other therapies targeting these genera identified in this study may be important for the prevention of thymoma.

The bacterial diversity tends to be higher in type A thymoma (less aggressive type) than in type B (more aggressive type). A study comparing the microbiota between tumor and normal peritumoral tissues in lung cancer demonstrated that the bacterial diversity was significantly higher in normal peritumoral tissues [6]. According to these data, cancer aggressiveness and alpha diversity are negatively correlated. Since the cancer microenvironment is more perturbed, dysbiosis might be enhanced; consequently, the bacterial diversity might decrease. In addition, because the lymphocyte counts in the tissues are higher in type B thymoma than in type A thymoma, the immunity against these bacteria may fundamentally differ between these types.

Using a LEfSe analysis, we identified variations in specific species between type A and B tumors and between tumors with and without the *GTF2I* mutation, indicating the differential microbiome functions in the development of each type of tumor [45]. We determined that Firmicutes, Bacilli, and Lactobacillales were common Gram-positive bacteria in type A thymoma, and Proteobacteria were common Gram-negative bacteria in type B thymoma. When the *p*-value based on the Kruskal–Wallis test was increased from 0.05 to 0.1 (Appendix A), 15 out of 20 bacteria that were significantly detected in the microbiota of type A thymoma were Gram-positive bacteria, and all four bacteria significantly detected in the microbiota of type B thymoma were Gram-negative bacteria. Although these findings indicated a correlation between the histological types (types A and B) and Gram-staining results for the microbiota, the biological significance of this correlation is unknown. Since Gram-negative bacteria are generally more pathogenic than Gram-positive bacteria, the former may be involved in carcinogenesis in type B thymoma, which is the more aggressive phenotype. Additionally, it is unclear from our observational study whether the identified bacterial differences are causally related to carcinogenesis or merely reflective of the disease process in thymoma. It is also difficult to practically prove how the microbiota colonized the thymoma tissue, which has no direct communication with the outside. In the future, detailed studies with a larger sample size may be needed.

We previously reported that the *GTF2I* mutation is a driver mutation in thymoma [46]. In the present study, specific species were identified between tumors with and without a *GTF2I* mutation. While Alphaproteobacteria were detected in a significantly high number of cases without the driver mutation in the *GTF2I* gene, a clear pathway involved in the oncological development of thymoma without a driver mutation remains to be demonstrated. Additionally, the mechanisms through which the microbiota in general contributes to carcinogenesis need to be examined in detail using a large sample size in the future.

This study is associated with some limitations. First, the patient cohort was relatively small owing to the rarity of the tumor. Second, patient survival could not be analyzed, as no patients have shown recurrence in the cohort. Third, no blood samples were analyzed for the microbiota containing the two genera, *Sphingomonas* and *Phenylobacterium*. An analysis of blood samples might have elucidated the reasons for the presence of the microbiota in the sterile anterior mediastinal environment [47]. In addition, the higher abundance of *Sphingomonas* and *Phenylobacterium* may be related to the impaired immunity of the tumor microenvironment, which may cause proliferation of these bacteria in the blood. Thus, they may be clinically applicable as serum biomarkers for thymoma. Fourth, a control thymus tissue should have been obtained to show that *Sphingomonas* and *Phenylobacterium* are microbiota associated with cancer progression. However, normal thymic tissue is known to rapidly atrophy and to be replaced with adipose tissue after puberty in the teens, and thymoma is presumed to be derived from atrophied residual thymic tissue. Therefore, even if a surgical specimen of adipose tissue in the anterior mediastinal of an age-matched population was obtained, thymic tissue is usually not left behind and cannot be analyzed. In addition, surgical specimens of the anterior mediastinal tissue of young individuals are extremely difficult to obtain, and it is ethically problematic to collect the functional thymic tissue of young individuals. Thus, it was not possible to compare microbiota between thymoma and normal thymic tissue in this study. In this context, a larger series of studies needs to be performed for evaluating the microbiome landscape of thymomas more comprehensively and elucidate the associations with clinical parameters through a more exhaustive multivariate analysis. Nevertheless, since the major aim of this preliminary analysis was identification of the thymoma-specific microbiota that should be prioritized for clinical development, the modestly sized samples provided useful insight.

## 4. Methods

### 4.1. Patients and Sample Preparation

In this study, we enrolled 19 patients in an unbiased manner who underwent surgical resection for thymoma at our hospital between January 2014 and August 2020. Since antibiotics would affect the microbiome, patients who had used oral or systematic antibiotics in the past 3 months were not included in this study. We obtained written informed consent for genetic research from all the patients in accordance with the protocols approved by the Institutional Review Board at Yamanashi Central Hospital. The specimens were categorized histologically according to the classification guidelines by the World Health Organization [48,49] and staged according to the Masaoka Staging System [19,50,51]. Sections of formalin-fixed and paraffin-embedded (FFPE) tissues were stained with hematoxylin–eosin and microdissected using the ArcturusXT laser-capture microdissection system (Thermo Fisher Scientific, Waltham, MA, USA), as previously reported [52,53,54,55,56,57]. For type AB thymomas, the type A and B portions were microdissected and examined separately. A thymoma is an encapsulated tumor, and the tumor tissue and surrounding fat are clearly separated by a fibrous capsule. In this study, DNA was extracted from the tumor tissue of the FFPE specimen of thymoma with laser-capture microdissection, so contamination of the adipose tissue around the tumor was unlikely.

We analyzed 22 samples obtained from all 19 patients, including three patients with type AB thymomas. The GeneRead DNA FFPE Kit (Qiagen, Hilden, Germany) was used according to the manufacturer’s instructions, and the DNA quality was evaluated using primers against ribonuclease P, as previously reported [58]. In the same manner, tumor DNA was extracted from FFPE samples obtained from patients with pancreatic cancer in our hospital (*n* = 30) and used as a control. As the preliminary experiment, PCR amplification of the 16S rDNA V4 region was attempted using distilled water, and a DNA elution buffer was used in the experiment for the samples, but DNA amplification was not obtained (below the detection sensitivity).

### 4.2. 16S rRNA Amplification and Targeted Sequencing

Although there is no hypervariable region of the 16S gene that allows an accurate classification of all bacterial strains at the domain to the species level, there is a known region that allows the near-perfect prediction at a specific taxonomic level [59]. In many studies on microbiome analyses, a commonly selected region is the V4 hypervariable region that allows a strain analysis at the phylum level with accuracy similar to that of the analysis of the complete 16S rRNA gene. The 16S rDNA V4 region was amplified using PCR and sequenced as described previously with minor modifications [7]. FFPE DNA was amplified using Platinum PCR SuperMix High Fidelity (Thermo Fisher Scientific) with the forward primer 5′-GTGYCAGCMGCCGCGGTAA-3′ (16S_rRNA_V4_515F) and reverse primer 5′-GGACTACNVGGGTWTCTAAT-3′ (16S_rRNA_V4_806R). The PCR products were confirmed using agarose gel electrophoresis and purified using Agencourt AMPure XP reagents (Beckman Coulter, Brea, CA, USA). End repair and barcode adaptors were ligated with the Ion Plus Fragment Library Kit (Thermo Fisher Scientific), in accordance with the manufacturer’s instructions, to construct the libraries. The library concentration was determined using an Ion Library Quantitation Kit (Thermo Fisher Scientific), and the same number of libraries was pooled for one sequence. Emulsion PCR and chip loading was performed on Ion Chef with the Ion PGM Hi-Q View Chef Kit; sequencing was performed on an Ion PGM Sequencer (Thermo Fisher Scientific). Sequence data were transferred to the IonReporter local server using the IonReporterUploader plugin. The coverage of the sequencing was 329.2 (Appendix A). Data was analyzed using the Metagenomics Research application using a custom primer set. The analytical parameters were set as the default. The control paraffin block without any tissue was processed similarly.

### 4.3. Data Analysis

The original raw tags were obtained through splicing the reads using FLASH (v 1.2.7) and subsequently filtered to acquire clean tags using QIIME (Version 1.9.1). To identify the taxa composition of each sample, the operational taxonomic units (OTUs) were classified on the effective tags with 97% identity using Usearch (Uparse v 7.0.1001) software. The presentative sequence of each OTU was annotated using the RDP classifier against the SILVA (SSU123)16S rRNA database using a confidence threshold of 80%, obtaining taxonomic classification at the phylum, class, order, family, genus, and species levels. Multiple sequence alignment was performed using MUSCLE3.6 (Version 3.8.31) to further explore the phylogenetic relationships among the different OTUs. The Shannon Index was performed using QIIME to determine the alpha diversity. Linear discriminant analysis (LDA) effect size (LEfSe) analyses were performed using the online LEfSe tool (http://huttenhower.sph.harvard.edu/lefse/ (accessed on 12 October 2020)). The LDA (linear discriminant analysis) threshold score was set at 2.

### 4.4. Targeted Deep Sequencing of GTF2I Mutation

In this study, the presence of a point mutation in the *GTF2I* gene was investigated in thymomas using targeted sequencing coupled with molecular barcoding, as we previously reported [47].

### 4.5. Immunohistochemistry for PD-L1

Specimens from the 19 patients were fixed using 10% buffered formalin. The formalin-fixed paraffin-embedded tissues were cut into 5-μm sections, deparaffinized, rehydrated, and stained in an automated system (Ventana Benchmark ULTRA system; Roche, Tucson, AZ, USA) using commercially available detection kits and antibodies against PD-L1 (28–8, ab205921; Abcam, Cambridge, MA, USA). PD-L1 was primarily localized to the cell membranes of the tumor cells, and its expression was determined quantitatively by two pathologists on the basis of the proportion of PD-L1-positive tumor cells. Cells were considered PD-L1-positive based on ≥1% PD-L1 expression.

### 4.6. Statistics

Continuous variables were presented as the mean ± standard deviation (SD) and compared using unpaired Student’s *t*-tests. One-way analysis of variance and the Tukey–Kramer multiple comparison test were used to detect significant differences between groups. *p*-values less than 0.05 in the two-tailed analyses were considered to denote statistical significance.

## 5. Conclusions

This is the first study that examined the microbiota in thymomas and revealed two genera specific to thymomas: *Sphingomonas* and *Phenylobacterium*.

## Figures and Tables

**Figure 1 jpm-11-01092-f001:**
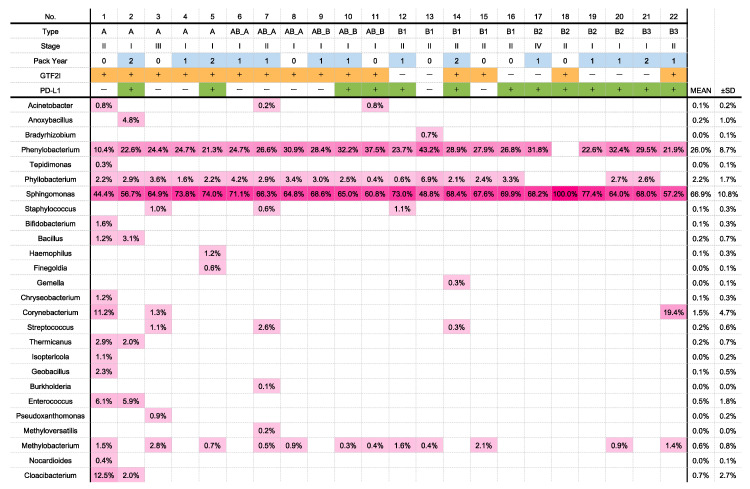
Composition and abundance of the dominant genera in all the samples. In total, 26 genera were identified. The heatmap visualizes the abundance of the detected genera. PY (pack-year) 0 represents nonsmokers; PY1, smokers with >0 to ≤30 packs/year history; and PY2, smokers with >30 packs/year history.

**Figure 2 jpm-11-01092-f002:**
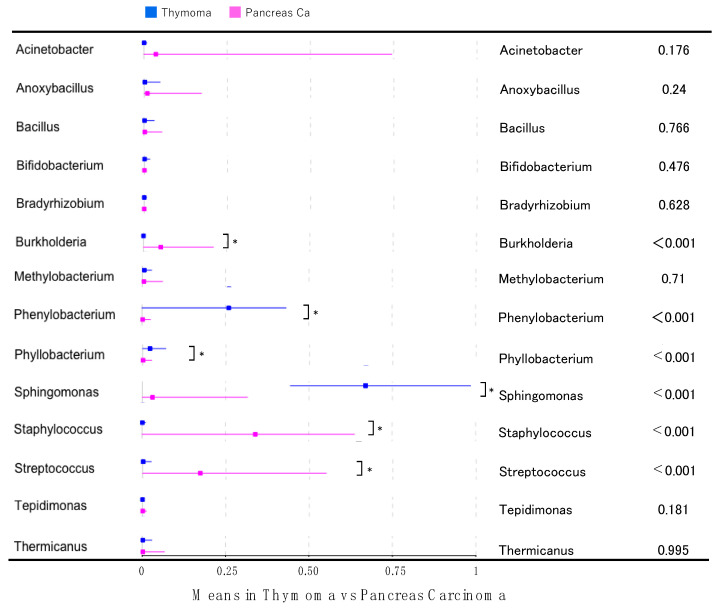
Microbiome differences between thymoma and pancreatic cancer samples. * *p* < 0.05.

**Figure 3 jpm-11-01092-f003:**
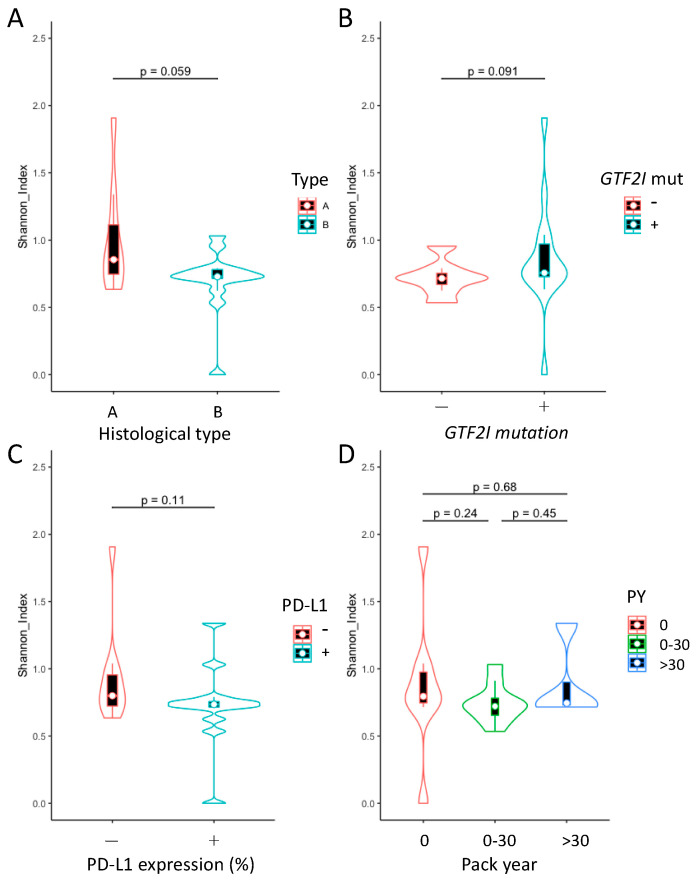
Taxonomic alpha diversity of thymoma microbiomes within samples in different groups. (**A**) Comparison of the Shannon Index between type A and B histology groups. (**B**) Comparison of the Shannon Index between tumors exhibiting the presence and those exhibiting an absence of the *GTF2I* driver mutation. (**C**) Comparison of the Shannon Index between tumors exhibiting the presence and those exhibiting an absence of PD-L1 expression on tumor cells. (**D**) Comparison of the Shannon Index among nonsmokers, light smokers, and heavy smokers. No significant difference was found among these groups.

**Figure 4 jpm-11-01092-f004:**
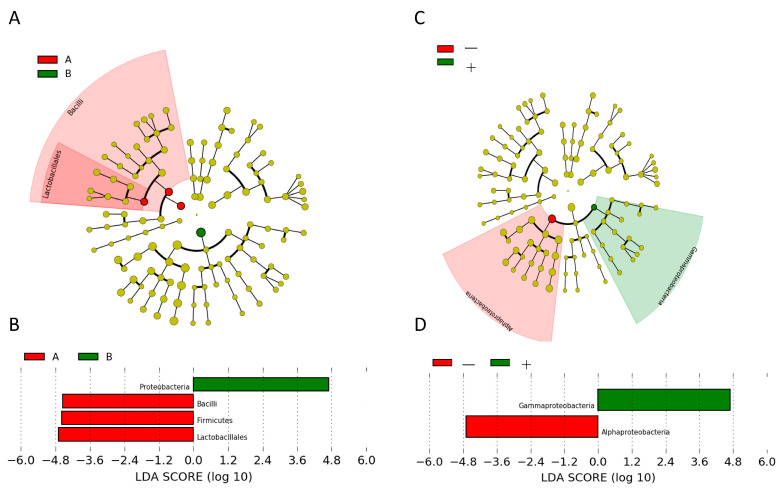
Differential taxa in the histology and driver mutation groups. (**A**) Cladogram of differential taxa between Types A and B histology. Dominant taxa are indicated in red for the type A group and in green for the type B group. (**B**) Kruskal–Wallis test results on the relative abundance between type A and type B histology. Type A is presented in the red column and type B in the green column. * *p* < 0.05. (**C**) The results of the LEfSe analysis between tumors exhibiting the presence and those exhibiting the absence of the *GTF2I* driver mutation. (**D**) Kruskal–Wallis test results for the relative abundance between tumors exhibiting the presence and those exhibiting the absence of the *GTF2I* driver mutation. * *p* < 0.05.

**Table 1 jpm-11-01092-t001:** Patient characteristics.

Parameter		Number of Patients	Overall Percentage
Total number		19	
Age, median (range)		68 (42–81)	
Sex			
	Male	11	57.9%
	Female	8	42.1%
Histology			
	Type A	5	26.3%
	Type AB	3	15.8%
	Type B1	5	26.3%
	Type B2	4	21.1%
	Type B3	2	10.5%
Tumor size (cm)			
	≤3	7	36.8%
	3 < size ≤ 5	8	42.1%
	>5	4	21.1%
Masaoka Stage			
	I	7	36.8%
	II	9	47.4%
	III	2	10.5%
	IV	1	5.3%
Smoking Status (Pack year)			
	0	7	36.8%
	0 < PY ≤ 30	8	42.1%
	>30	4	21.1%
Myasthenia gravis			
	present	1	5.3%
	absent	18	94.7%

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
