# Peer review of "Sphingomonas and Phenylobacterium as Major Microbiota in Thymic Epithelial Tumors"

_jpm, 2021, doi:10.3390/jpm11111092_

Round 1

Reviewer 1 Report

The authors of the current manuscript provide interesting and unique findings that human thymomas isolated from 22 patients have evidence of two bacterial species based on PCR using 16s rRNA probes. These two species, Sphingomonas and Phenylobacteria are evident in the thymomas and independent of the grade of thymoma. More importantly, they establish that such a pattern in not seen in pancreatic tumors nor in normal blood samples. While these results are interesting, there is a lack of sufficient detailed information needed to support the observations.

Critical issues.

  1. Was any background control such as an elution buffer included to profile the potential contaminants in the DNA extraction step. This is also important for the tissue microbiome analysis, as swabs from the microtome and blade used to cut the sections are recommended to profile potential contaminants. The present study does not seem to include any of those
  1. There was no supplementary file provided so the key information regarding the 16S analyses were not available
  2. The authors never establish whether the read count confirms the presence of the indicated species. What threshold of sequencing read depth was used to distinguish between noise and a signal? Thus, it remains possible the read count is below what is considered true reads. OTU call in control and patient tissues would help address this
  3. Have the authors performed deep sequencing reads to determine whether the signal they report corresponds to an intact bacterial genome as opposed to just the stable 16s rRNA sequence
  4. An important control for the study is normal thymus tissue from, if possible, age matched tissues. This could come from post-mortem tthymus issues if needed, or alternatively, patients undergoing cardiothoracic surgery or patients with myasthenia gravis, wherein the thymus is removed. A control thymus is needed to verify that the Sphingo and Phenylo signals are reflective of cancer progression
  5. Is it possible that the detection of the species as indicated, comes from DCs, neutrophils and/or or macrophages that have accumulated in the tissue due to either the vascularization of the tissue or changes in the cell types present
  6. In the context of question 5, could it be that the signature noted comes from adipose tissue that has accumulated throughout the thymus? This is often noted in thymus tissue from older patients and may not at all be a signature related to the thymoma itself

Minor comment

The authors state “Sphingomonas is a bacterial genus that was separated from Pseudomonas approximately 30 years ago” Should this read 30 million years ago?

Author Response

Critical issues.

  1. Was any background control such as an elution buffer included to profile the potential contaminants in the DNA extraction step. This is also important for the tissue microbiome analysis, as swabs from the microtome and blade used to cut the sections are recommended to profile potential contaminants. The present study does not seem to include any of those.

Response: The control paraffin block without any tissue was processed similarly as the thymoma samples, but no OTU was found in those negative control samples. In addition, as a preliminary experiment, PCR amplification of 16S rDNA V4 region was attempted using distilled water and DNA elution buffer used in the experiment as samples, but DNA amplification was not obtained (below the detection sensitivity). Therefore, as a prerequisite for the experiment, contamination from lab equipment and chemicals is considered to be negative.

We added these descriptions in the Methods and Results sections.

  1. There was no supplementary file provided so the key information regarding the 16S analyses were not available

Response: According to the reviewer’s suggestion, we added the supplementary file regarding 16S rRNA analysis (Supplementary Table 2).

  1. The authors never establish whether the read count confirms the presence of the indicated species. What threshold of sequencing read depth was used to distinguish between noise and a signal? Thus, it remains possible the read count is below what is considered true reads. OTU call in control and patient tissues would help address this

Response: Coverage of the sequencing was described in the Methods section, and Supplementary Table 1 was added to show the coverage of each sample.

  1. Have the authors performed deep sequencing reads to determine whether the signal they report corresponds to an intact bacterial genome as opposed to just the stable 16s rRNA sequence

Response: The coverage of the sequencing was 329.2 (>30), thus deep targeted sequencing was performed in this study. Moreover, we followed the similar methods and pipelines as previously reported for microbiome analysis using FFPE tissue samples (Riquelme et al. Cell 2019, 178 (4), 795-806).

Therefore, we are confident that microbiota we identified through the analysis corresponds to an intact bacterial genome.

  1. An important control for the study is normal thymus tissue from, if possible, age matched tissues. This could come from post-mortem thymus issues if needed, or alternatively, patients undergoing cardiothoracic surgery or patients with myasthenia gravis, wherein the thymus is removed. A control thymus is needed to verify that the Sphingo and Phenylo signals are reflective of cancer progression

Response:

We agree with the reviewer’s comments. A control thymus tissue should be obtained to show that Sphingomonas and Phenylobacterium are microbiota associated with cancer progression. However, normal thymic tissue is known to rapidly atrophy and to be replaced with adipose tissue after puberty in the teens, and thymoma is presumed to be derived from atrophied residual thymic tissue. Therefore, even if a surgical specimen of adipose tissue in the anterior mediastinal of the age-matched population is obtained, the thymic tissue is usually not left behind and cannot be analyzed. In addition, surgical specimens of the anterior mediastinal tissue of young individuals are extremely difficult to obtain, and it is ethically problematic to collect functional thymic tissue of young individuals. Thus, it was not possible to compare microbiota between thymoma and normal thymic tissue in this study because it is practically difficult to collect normal thymic tissue. We added these limitations of the study in the Discussion section.

Furthermore, at this time, there is little evidence that Sphingomonas and Phenylobacterium are microbiota involved in thymoma progression, so the related description in the paper has been changed or deleted.

  1. Is it possible that the detection of the species as indicated, comes from DCs, neutrophils and/or or macrophages that have accumulated in the tissue due to either the vascularization of the tissue or changes in the cell types present

Response: Inflammatory cells and immunocompetent cells are found in the tumor microenvironment, so it is difficult to practically prove how the microbiota colonized the thymoma tissue and whether the microbiota was the cause or the result of oncogenesis.

We added these descriptions in the Discussion sections.

Verifying how and when the two strains colonized tissues such as thymus, which has no direct communication with the outside, could be beyond the scope of this study and thus should be the subject of next research.

  1. In the context of question 5, could it be that the signature noted comes from adipose tissue that has accumulated throughout the thymus? This is often noted in thymus tissue from older patients and may not at all be a signature related to the thymoma itself

Response: A thymoma is an encapsulated tumor, which is rare in the human body, and the tumor tissue and surrounding fat are clearly separated by a fibrous capsule. In this study, DNA was extracted from the tumor tissue of the FFPE specimen of thymoma with laser-capture microdissection, so contamination of the adipose tissue around the tumor is unlikely.

We added these descriptions in the Methods section.

Minor comment

The authors state “Sphingomonas is a bacterial genus that was separated from Pseudomonas approximately 30 years ago” Should this read 30 million years ago?

Response: We are sorry for this misleading expression and replaced “separated” with “subclassified” in that sentence.

“Sphingomonas is a bacterial genus that was subclassified from Pseudomonas approximately 30 years ago.”

Thank you for your thoughtful comments.

Reviewer 2 Report

I would like to congratulate the author of an interesting article on microbiota in thymoma.

In a study conducted on 19 patients who underwent surgery for thymoma, with the use of 16s RNA sequence and metagenomic analyses, the authors identified Sphingomonas and Phenylobacterium genera in to be abundant in tumor tissue. In addition, the authors found tendency toward increased microbiome diversity in type A compared to type B thymoma.

This is probably the first study to attempt to identify the microbiome in thymomas. Despite searching the literature with the use of appropriate phrases using most of the available search engines (Pubmed, Webofknowledge, Ovid, Scopus), I was unable to find a study in which this topic would be discussed. From this point of view, I believe that the work should be published in the Journal.

Apart from its novelty, which I mentioned above, the manuscript is written in good quality English and its layout fully complies with the guidelines of the Journal. Background and aims are described in the introduction, the methodology is correct, the results are clearly presented, and the discussion refers to the current literature.

I have only a few comments.

  1. On the basis of the study conducted by the authors, we can draw conclusions only about the existence of a relation between microbiota and the tumor, but not about the existence of cause-and-effect relationships between them. Moreover, the existence of a cause-and-effect relationship between the microbiome and cancer pathogenesis in general has been not proven (yet) in most types of neoplasms. Hence, discussions on the influence of the microbiome on the development of thymoma, as well as the possibility of preventing thymoma by the use of antibiotics are purely hypothetical and should be limited to the “Discussion” section at best. Taking this into account, I would suggest the following changes:
    1. Abstract: change the sentence “These genera appear to comprise thymoma-specific microbiota involved in tumor progression; thus, they could serve as targets for the prevention of thymoma” to: “These genera appear to comprise thymoma-specific microbiota”
    2. Introduction: change the sentence “Although recent reports indicate the association between the microbiome and the development of colorectal, oral, pancreatic, lung, and other cancers [22-27], there is no report on the involvement of the microbiome in thymoma” to: “Although recent reports indicate the association between the microbiome and colorectal, oral, pancreatic, lung, and other cancers [22-27], there is no report on the microbiome in thymoma”
    3. Discussion: change “… [44]. This study indicates that the microbiota may be associated with the development of thymoma” to: “… [44]. This study indicates that the microbiota may be associated with thymoma”
    4. Conclusions: delete all following sentences, because the are not based on the results of the study: “The development of probiotic models for controlling these two genera may enable the prevention of thymoma. Particularly, antibiotic and vaccine therapies for Sphingomonas and Phenylobacterium are expected to be clinically applicable for preventing thymoma in patients with myasthenia gravis who are likely to develop thymoma.”
  2. The number of patients with particular sub-types of thymoma was very small, moreover, no statistically significant difference was found between types A and B in terms of microbiome differentiation, but only a tendency (p=0.059). From this point of view, the statement that “…type A thymoma exhibited higher bacterial diversity than type B thymoma” is an overinterpretation of the obtained results. I suggest to delete this phrase, or substantially change it (to include word “tendency” or “tended to”).
  3. A topic that has hardly been addressed in the case of research on microbiota in tissues and organs previously considered sterile is the possibility of tissue contamination during or after surgery. I am curious about the authors' opinions on this subject.
  4. In order to identify thymoma-specific microbiota, the authors compared the microbiota of thymoma to the microbiota of pancreatic cancer. However, there is a theoretical possibility that the microbiota, interpreted by the authors as specific for thymoma, is in fact a microbiota characteristic of the thymus, both healthy and diseased. To exclude this possibility, the thymoma microbiota should be compared with that of the thymus in healthy patients in another study. For the purposes of the current manuscript, however, I would suggest adding a short comments on these issues to the “Discussion” and “Limitations” sections.
  5. In some parts of the discussion the authors create somewhat too far-reaching and overly enthusiastic hypotheses. Nevertheless, the discussion as a whole is interesting and thought-provoking and in my opinion does not require changes beyond those mentioned in the previous points.

Overall, I find the study innovative and the manuscripts written in an interesting way. It is highly likely, that authors of the future studies on this and related topics will find this study interesting.

I can recommend the manuscript for the publication in the Journal of Personalized Medicine after minor revision.

Author Response

I have only a few comments.

  1. On the basis of the study conducted by the authors, we can draw conclusions only about the existence of a relation between microbiota and the tumor, but not about the existence of cause-and-effect relationships between them. Moreover, the existence of a cause-and-effect relationship between the microbiome and cancer pathogenesis in general has been not proven (yet) in most types of neoplasms. Hence, discussions on the influence of the microbiome on the development of thymoma, as well as the possibility of preventing thymoma by the use of antibiotics are purely hypothetical and should be limited to the “Discussion” section at best. Taking this into account, I would suggest the following changes:
    1. Abstract: change the sentence “These genera appear to comprise thymoma-specific microbiota involved in tumor progression; thus, they could serve as targets for the prevention of thymoma” to: “These genera appear to comprise thymoma-specific microbiota”
    2. Introduction: change the sentence “Although recent reports indicate the association between the microbiome and the development of colorectal, oral, pancreatic, lung, and other cancers [22-27], there is no report on the involvement of the microbiome in thymoma” to: “Although recent reports indicate the association between the microbiome and colorectal, oral, pancreatic, lung, and other cancers [22-27], there is no report on the microbiome in thymoma”
    3. Discussion: change “… [44]. This study indicates that the microbiota may be associated with the development of thymoma” to: “… [44]. This study indicates that the microbiota may be associated with thymoma”
    4. Conclusions: delete all following sentences, because the are not based on the results of the study: “The development of probiotic models for controlling these two genera may enable the prevention of thymoma. Particularly, antibiotic and vaccine therapies for Sphingomonas and Phenylobacterium are expected to be clinically applicable for preventing thymoma in patients with myasthenia gravis who are likely to develop thymoma.”

Response: We totally agree with the reviewer, and revised the manuscript as the reviewer suggested.

  1. The number of patients with particular sub-types of thymoma was very small, moreover, no statistically significant difference was found between types A and B in terms of microbiome differentiation, but only a tendency (p=0.059). From this point of view, the statement that “…type A thymoma exhibited higher bacterial diversity than type B thymoma” is an overinterpretation of the obtained results. I suggest to delete this phrase, or substantially change it (to include word “tendency” or “tended to”).

Response: We agree with the reviewer. As the reviewer suggested, the sentence in the abstract was revised, by adding “tended to”.

“Type A thymoma tended to exhibit higher bacterial diversity than type B thymoma.”

  1. A topic that has hardly been addressed in the case of research on microbiota in tissues and organs previously considered sterile is the possibility of tissue contamination during or after surgery. I am curious about the authors' opinions on this subject.

Response: I agree with the reviewer. The control paraffin block without any tissue was processed similarly as the thymoma samples, but no OTU was found in those negative control samples. In addition, as a preliminary experiment, PCR amplification of 16S rDNA V4 region was attempted using distilled water and DNA elution buffer used in the experiment as samples, but DNA amplification was not obtained (below the detection sensitivity). Therefore, as a prerequisite for the experiment, contamination from lab equipment and chemicals is considered to be negative.

We added these descriptions in the Methods and Results sections.

  1. In order to identify thymoma-specific microbiota, the authors compared the microbiota of thymoma to the microbiota of pancreatic cancer. However, there is a theoretical possibility that the microbiota, interpreted by the authors as specific for thymoma, is in fact a microbiota characteristic of the thymus, both healthy and diseased. To exclude this possibility, the thymoma microbiota should be compared with that of the thymus in healthy patients in another study. For the purposes of the current manuscript, however, I would suggest adding a short comments on these issues to the “Discussion” and “Limitations” sections.

Response:

We agree with the reviewer’s comments. A control thymus tissue should be obtained to show that Sphingomonas and Phenylobacterium are microbiota associated with cancer progression. However, normal thymic tissue is known to rapidly atrophy and to be replaced with adipose tissue after puberty in the teens, and thymoma is presumed to be derived from atrophied residual thymic tissue. Therefore, even if a surgical specimen of adipose tissue in the anterior mediastinal of the age-matched population is obtained, the thymic tissue is usually not left behind and cannot be analyzed. In addition, surgical specimens of the anterior mediastinal tissue of young individuals are extremely difficult to obtain, and it is ethically problematic to collect functional thymic tissue of young individuals. Thus, it was not possible to compare microbiota between thymoma and normal thymic tissue in this study because it is practically difficult to collect normal thymic tissue. We added these limitations of the study in the Discussion section.

  1. In some parts of the discussion the authors create somewhat too far-reaching and overly enthusiastic hypotheses. Nevertheless, the discussion as a whole is interesting and thought-provoking and in my opinion does not require changes beyond those mentioned in the previous points.

Response: Thank you for your thoughtful comments.

I have only a few comments.

  1. On the basis of the study conducted by the authors, we can draw conclusions only about the existence of a relation between microbiota and the tumor, but not about the existence of cause-and-effect relationships between them. Moreover, the existence of a cause-and-effect relationship between the microbiome and cancer pathogenesis in general has been not proven (yet) in most types of neoplasms. Hence, discussions on the influence of the microbiome on the development of thymoma, as well as the possibility of preventing thymoma by the use of antibiotics are purely hypothetical and should be limited to the “Discussion” section at best. Taking this into account, I would suggest the following changes:
    1. Abstract: change the sentence “These genera appear to comprise thymoma-specific microbiota involved in tumor progression; thus, they could serve as targets for the prevention of thymoma” to: “These genera appear to comprise thymoma-specific microbiota”
    2. Introduction: change the sentence “Although recent reports indicate the association between the microbiome and the development of colorectal, oral, pancreatic, lung, and other cancers [22-27], there is no report on the involvement of the microbiome in thymoma” to: “Although recent reports indicate the association between the microbiome and colorectal, oral, pancreatic, lung, and other cancers [22-27], there is no report on the microbiome in thymoma”
    3. Discussion: change “… [44]. This study indicates that the microbiota may be associated with the development of thymoma” to: “… [44]. This study indicates that the microbiota may be associated with thymoma”
    4. Conclusions: delete all following sentences, because the are not based on the results of the study: “The development of probiotic models for controlling these two genera may enable the prevention of thymoma. Particularly, antibiotic and vaccine therapies for Sphingomonas and Phenylobacterium are expected to be clinically applicable for preventing thymoma in patients with myasthenia gravis who are likely to develop thymoma.”

Response: We totally agree with the reviewer, and revised the manuscript as the reviewer suggested.

  1. The number of patients with particular sub-types of thymoma was very small, moreover, no statistically significant difference was found between types A and B in terms of microbiome differentiation, but only a tendency (p=0.059). From this point of view, the statement that “…type A thymoma exhibited higher bacterial diversity than type B thymoma” is an overinterpretation of the obtained results. I suggest to delete this phrase, or substantially change it (to include word “tendency” or “tended to”).

Response: We agree with the reviewer. As the reviewer suggested, the sentence in the abstract was revised, by adding “tended to”.

“Type A thymoma tended to exhibit higher bacterial diversity than type B thymoma.”

  1. A topic that has hardly been addressed in the case of research on microbiota in tissues and organs previously considered sterile is the possibility of tissue contamination during or after surgery. I am curious about the authors' opinions on this subject.

Response: I agree with the reviewer. The control paraffin block without any tissue was processed similarly as the thymoma samples, but no OTU was found in those negative control samples. In addition, as a preliminary experiment, PCR amplification of 16S rDNA V4 region was attempted using distilled water and DNA elution buffer used in the experiment as samples, but DNA amplification was not obtained (below the detection sensitivity). Therefore, as a prerequisite for the experiment, contamination from lab equipment and chemicals is considered to be negative.

We added these descriptions in the Methods and Results sections.

  1. In order to identify thymoma-specific microbiota, the authors compared the microbiota of thymoma to the microbiota of pancreatic cancer. However, there is a theoretical possibility that the microbiota, interpreted by the authors as specific for thymoma, is in fact a microbiota characteristic of the thymus, both healthy and diseased. To exclude this possibility, the thymoma microbiota should be compared with that of the thymus in healthy patients in another study. For the purposes of the current manuscript, however, I would suggest adding a short comments on these issues to the “Discussion” and “Limitations” sections.

Response:

We agree with the reviewer’s comments. A control thymus tissue should be obtained to show that Sphingomonas and Phenylobacterium are microbiota associated with cancer progression. However, normal thymic tissue is known to rapidly atrophy and to be replaced with adipose tissue after puberty in the teens, and thymoma is presumed to be derived from atrophied residual thymic tissue. Therefore, even if a surgical specimen of adipose tissue in the anterior mediastinal of the age-matched population is obtained, the thymic tissue is usually not left behind and cannot be analyzed. In addition, surgical specimens of the anterior mediastinal tissue of young individuals are extremely difficult to obtain, and it is ethically problematic to collect functional thymic tissue of young individuals. Thus, it was not possible to compare microbiota between thymoma and normal thymic tissue in this study because it is practically difficult to collect normal thymic tissue. We added these limitations of the study in the Discussion section.

  1. In some parts of the discussion the authors create somewhat too far-reaching and overly enthusiastic hypotheses. Nevertheless, the discussion as a whole is interesting and thought-provoking and in my opinion does not require changes beyond those mentioned in the previous points.

Response: Thank you for your thoughtful comments.

I have only a few comments.

  1. On the basis of the study conducted by the authors, we can draw conclusions only about the existence of a relation between microbiota and the tumor, but not about the existence of cause-and-effect relationships between them. Moreover, the existence of a cause-and-effect relationship between the microbiome and cancer pathogenesis in general has been not proven (yet) in most types of neoplasms. Hence, discussions on the influence of the microbiome on the development of thymoma, as well as the possibility of preventing thymoma by the use of antibiotics are purely hypothetical and should be limited to the “Discussion” section at best. Taking this into account, I would suggest the following changes:
    1. Abstract: change the sentence “These genera appear to comprise thymoma-specific microbiota involved in tumor progression; thus, they could serve as targets for the prevention of thymoma” to: “These genera appear to comprise thymoma-specific microbiota”
    2. Introduction: change the sentence “Although recent reports indicate the association between the microbiome and the development of colorectal, oral, pancreatic, lung, and other cancers [22-27], there is no report on the involvement of the microbiome in thymoma” to: “Although recent reports indicate the association between the microbiome and colorectal, oral, pancreatic, lung, and other cancers [22-27], there is no report on the microbiome in thymoma”
    3. Discussion: change “… [44]. This study indicates that the microbiota may be associated with the development of thymoma” to: “… [44]. This study indicates that the microbiota may be associated with thymoma”
    4. Conclusions: delete all following sentences, because the are not based on the results of the study: “The development of probiotic models for controlling these two genera may enable the prevention of thymoma. Particularly, antibiotic and vaccine therapies for Sphingomonas and Phenylobacterium are expected to be clinically applicable for preventing thymoma in patients with myasthenia gravis who are likely to develop thymoma.”

Response: We totally agree with the reviewer, and revised the manuscript as the reviewer suggested.

  1. The number of patients with particular sub-types of thymoma was very small, moreover, no statistically significant difference was found between types A and B in terms of microbiome differentiation, but only a tendency (p=0.059). From this point of view, the statement that “…type A thymoma exhibited higher bacterial diversity than type B thymoma” is an overinterpretation of the obtained results. I suggest to delete this phrase, or substantially change it (to include word “tendency” or “tended to”).

Response: We agree with the reviewer. As the reviewer suggested, the sentence in the abstract was revised, by adding “tended to”.

“Type A thymoma tended to exhibit higher bacterial diversity than type B thymoma.”

  1. A topic that has hardly been addressed in the case of research on microbiota in tissues and organs previously considered sterile is the possibility of tissue contamination during or after surgery. I am curious about the authors' opinions on this subject.

Response: I agree with the reviewer. The control paraffin block without any tissue was processed similarly as the thymoma samples, but no OTU was found in those negative control samples. In addition, as a preliminary experiment, PCR amplification of 16S rDNA V4 region was attempted using distilled water and DNA elution buffer used in the experiment as samples, but DNA amplification was not obtained (below the detection sensitivity). Therefore, as a prerequisite for the experiment, contamination from lab equipment and chemicals is considered to be negative.

We added these descriptions in the Methods and Results sections.

  1. In order to identify thymoma-specific microbiota, the authors compared the microbiota of thymoma to the microbiota of pancreatic cancer. However, there is a theoretical possibility that the microbiota, interpreted by the authors as specific for thymoma, is in fact a microbiota characteristic of the thymus, both healthy and diseased. To exclude this possibility, the thymoma microbiota should be compared with that of the thymus in healthy patients in another study. For the purposes of the current manuscript, however, I would suggest adding a short comments on these issues to the “Discussion” and “Limitations” sections.

Response:

We agree with the reviewer’s comments. A control thymus tissue should be obtained to show that Sphingomonas and Phenylobacterium are microbiota associated with cancer progression. However, normal thymic tissue is known to rapidly atrophy and to be replaced with adipose tissue after puberty in the teens, and thymoma is presumed to be derived from atrophied residual thymic tissue. Therefore, even if a surgical specimen of adipose tissue in the anterior mediastinal of the age-matched population is obtained, the thymic tissue is usually not left behind and cannot be analyzed. In addition, surgical specimens of the anterior mediastinal tissue of young individuals are extremely difficult to obtain, and it is ethically problematic to collect functional thymic tissue of young individuals. Thus, it was not possible to compare microbiota between thymoma and normal thymic tissue in this study because it is practically difficult to collect normal thymic tissue. We added these limitations of the study in the Discussion section.

  1. In some parts of the discussion the authors create somewhat too far-reaching and overly enthusiastic hypotheses. Nevertheless, the discussion as a whole is interesting and thought-provoking and in my opinion does not require changes beyond those mentioned in the previous points.

Response: Thank you for your thoughtful comments.